# Research on Measuring Pure Membrane Electrical Resistance under the Effects of Salinity Gradients and Diffusion Boundary Layer and Double Layer Resistances

**DOI:** 10.3390/membranes12080816

**Published:** 2022-08-22

**Authors:** Yang Zhao, Liang Duan

**Affiliations:** 1State Key Joint Laboratory of Environment Simulation and Pollution Control, School of Environment, Tsinghua University, Beijing 100084, China; 2Chinese Research Academy of Environmental Sciences, Beijing 100012, China

**Keywords:** pure membrane electrical resistance, Luggin capillary coupled with AC impedance spectroscopy (EIS), salinity gradients, circulation velocity

## Abstract

Forward osmosis membranes are an emerging technology with great potential applicability in energy-efficient wastewater treatments and the differentiation between two solutions. Such solutions often differ in their concentrations or compositions. In this study, the membrane electrical resistances of three different membranes, including cation or anion-exchange membranes and forward osmosis membranes, were analyzed by Luggin capillary coupled with AC impedance spectroscopy (EIS) so as to obtain the real membrane and ion transfer impedance values near the membrane interface. The results reveal that the membrane impedance obtained by both the DC and AC approaches decreased as the lowest external solution concentration increased. Furthermore, the relationship between the membrane conductivity and the internal salt solution concentration was also investigated. It can be seen that the external ion concentration is directly proportional to the free ion concentration in the membrane, and the free ion concentration in the membrane is closely related to the membrane electrical resistance.

## 1. Introduction

Ion-exchange membranes are widely used in the fields of electrodialysis, Donnan separation, diffusion dialysis, fuel cells, and others. Due to the fixed cation and anion functional groups present on the membrane surface, ion-exchange membranes are able to transfer both cations and anions. The functional groups on the membrane’s surface also influence its characteristics, especially its electrical resistance, and affect the efficiency of the power production of microbial fuel cells directly [1,2,3,4]. The key problem that restricts electrochemical technologies, including microbial fuel cells, is the high internal electrical resistance, and the separator is an integral part of microbial fuel cells. Especially in the case of the salt concentration gradient, there is little research on the change law of membrane electrical resistance. Therefore, studying the measurement method of membrane electrical resistance plays an important role in exploring and improving the power production capacity of microbial fuel cell (MFCs) and putting it into practical application.

Membrane properties and especially membrane electrical resistance have great influences on the power output obtainable in the microbial fuel cell. Dlugolecki et al. used the DC method to determine the membrane electrical resistance of ion-exchange membranes as a function of the solution concentration and observed a significant increase in membrane electrical resistance below 0.1 M NaCl [5]. However, DC methods not only measure the pure membrane electrical resistance, but also include diffusion boundary layer and double layer effects. Therefore, the authors were not able to attribute the increase in the electrical resistance to one of these effects, because they used the DC method [5,6] and, due to its limitations, the DC method cannot separately identify the individual membrane electrical resistance, the diffusion boundary layer impedance, and the double electric layer impedance; it can only derive their sum. As a result, identifying a method that can be used to accurately measure the membrane electrical resistance is important when analyzing the variations in the pattern of the membrane impedance under different conditions, as well as the means by which the membrane electrical resistance can be reduced.

Permeability coefficients and hydraulic resistance in membrane processes, such as RO, FO, and PRO, are also often used to describe membrane resistance [6,7,8,9]. The long-term treatment of wastewater with high pollution and complex components still causes serious membrane pollution [10]. The pollution layer formed on the membrane surface increases the hydraulic resistance [11,12,13], resulting in the decrease in the membrane water flux [14,15,16]. The pollutants deposited on the fouling membrane surface block the membrane pores and change the rejection rate of the pollutants and solutes [17,18,19]. Differently from the hydraulic resistance, the salt flux showed an upward trend after membrane fouling. Zhao et al. found that the salt mass transfer resistance system was not significantly increased by the membrane pollution after fouling, resulting in the reverse salt flux being increased to 2.3 times that of the fresh membrane, which was far higher than the impact of the water flux decrease caused by the membrane fouling, thus reducing the membrane electrical resistance [20]. Therefore, the hydraulic resistance of a single membrane cannot accurately reflect the trend of the change in the electrochemical resistance of the membrane in actual operation and its relationship with the operating conditions.

AC impedance spectroscopy (EIS) can effectively solve the problems with DC chronopotentiometry and the hydraulic resistance measuring method, because different impedance values have different effects on the applied current at different frequencies. It can thus distinguish between the membrane electrical resistance itself, the double electric layer impedance, and diffusion layer impedance [21]. When a current passes through the membrane, the common impedance of both the solution and the membrane can be calculated by measuring the voltage on both sides of the membrane, according to Ohm’s law. The membrane electrical resistance can then be obtained through a test of the blank group [22]. However, when the current passes through the ion-exchange membrane, the ions with a charge opposite to that of the surface of the membrane usually function as the carriers of the charge transfer and deliver the charge through the membrane. Meanwhile, in the main solution, the charge can be transferred through both cations and anions, so there will be a difference in the number of ions transferred between the membrane and the main solution, resulting in the production of a diffusion boundary layer [23]. At the same time, when ions are transferred to the membrane surface, they are also affected by the double electric layer’s impedance, which forms on the membrane’s surface [24]. Both the double electric layer impedance and the diffusion boundary layer will hinder the ion transfer, especially under the conditions of low salt concentrations, which some researchers have shown to be a condition in which the impedance of the boundary layer has a great influence [25].

In this article, the membrane electrical resistances of three different membranes—cation- and anion-exchange membranes and forward osmosis membranes—are analyzed by Luggin capillary coupled with AC impedance spectroscopy (EIS), in order to obtain the real membrane impedance and ion transfer impedance near the membrane interface, including the impedance of the double electric layer and the diffusion layer. In addition, we analyze different salt solubility and circulation velocity conditions. This approach solves the problem whereby the traditional DC method cannot precisely distinguish between the real membrane electrical resistance and additional impedance. It will also play a key role in further research on the accurate measurement of the membrane electrical resistance and analyses of the mechanisms of high-efficiency production in a forward osmosis microbial fuel cell system [26].

## 2. Materials and Methods

The membrane electrical resistance was measured by chronopotentiometry and AC impedance spectroscopy. The experimental device diagram is shown below (Figure 1).

The reactor consisted of two chambers, each with a volume of 2 dm^3^, and the membrane’s size was 2.835 cm^2^. Before the test, it was necessary to soak the membrane being tested in a solution with the corresponding concentration for 24 h. The relatively small membrane area for measurement was selected to prevent the bulging deformation of the membrane and reduce the current density passing through the Ag/AgCl reference electrode. NaCl solutions with concentrations of 0.05 M and 0.5 M were used for the electrode solution, which was circulated in the two chambers at a specific flow rate using a peristaltic pump, and the circulation velocity was set as 0.1 L/min, 0.4 L/min, or 0.8 L/min. The solution was kept in a water bath at a constant temperature of 25 °C, and the concentration of the solution was monitored using a conductivity meter. The chronopotentiometry and AC impedance spectroscopy were completed at the electrochemical workstation. The potential difference and membrane impedance on both sides of the membrane were obtained by measuring the voltage between the reference electrodes. The reference electrode, whose diameter was 6 mm, was placed in the liquid storage chamber connected to the Luggin capillary (the diameter of the pipe was 6.5 mm), and the type and concentration of the solution in the capillary were consistent with those of the solution in the chamber.

The cation- and anion-exchange membranes were selected as the research objects. As regards the limiting of the current, the potential difference between the two sides of the membrane with different current densities was measured with the intention of calculating the membrane electrical resistance. When a low concentration of salt is utilized (i.e., 0.05 M NaCl), the limiting current varies based on the type of membrane, ranging from 2 to 6 mA, according to the literature [27]. Consequently, the current gradient was set as 1 mA, 1.5 mA, 2 mA, 2.5 mA, 3 mA, 3.5 mA, 4 mA, 4.5 mA, and 5 mA. For the 0.5 M NaCl, the limiting current was over 100 mA. Therefore, the current gradient selected increased by 10 mA every one minute, starting from 10 mA and ending when it reached 100 mA.

In the chronopotentiometric experiment, the membrane electrical resistance was determined via a slope, in which the current density (mA/cm^2^) was set as the *x*-axis and the potential difference (mV) was set as the *y*-axis. The obtained membrane electrical resistance R_m+s_ was determined by the impedance of the membrane itself and the impedance of the solution [28]:R_m+s_ = U/i(1)
where R_m+s_ is the sum of the impedance values of the membrane and the solution (Ω m^2^), U is the voltage difference between the reference electrodes (V), and i is the current density (A/m^2^). To obtain the impedance of the membrane itself R_m_, the impedance of the solution Rs needs to be subtracted. Consequently, a blank experiment is required, in which the impedance of the system Rs is measured after the membrane to be measured is removed from the device [29].

In the high-frequency area, if the phase angle of the voltage and current is zero, and the equivalent circuit diagram is characterized by individual electrical resistance, it can be concluded that the membrane electrical resistance (including the solution impedance) is the impedance value of the real part at the first intersection of the semicircle and the real part. The membrane electrical resistance in this experiment is given by the voltage difference between the two sides of the membrane to be measured, which is the same as the electrical resistance in the equivalent circuit. In the medium-frequency region, the double electric layer’s impedance can be obtained [30]. The equivalent circuit is represented by the parallel connection of a resistor R_DL_ and a capacitor C, which is the difference between the total impedance and the membrane impedance at the intersection of the second semicircle and the real part. In the low-frequency region, the diffusion layer’s impedance can be obtained. The equivalent circuit is also represented by the parallel connection of a resistor R_DBL_ and the constant phase angle element Q, which comprises the difference between the impedance at the first intersection of the second semicircle and the real part [31]. Physically speaking, the constant phase angle element Q is a non-ideal capacitor, formed by the diffusion boundary layer, and its size can be calculated by applying the angular velocity at the apex of a semicircular arc.

In AC impedance spectroscopy, the response voltage of the sinusoidal AC on both sides of the membrane to be measured is obtained by measuring the voltage between the Ag/AgCl reference electrodes. The AC frequency is set at 103 HZ to 10.3 HZ, and the amplitude of the disturbed sine wave’s current is 3 mA. The impedance data are incorporated into the Nyquist diagram through the equivalent circuit described above; the Nyquist diagram reflects the real part Re (Z) and the imaginary part lm (Z) of the impedance data. The fitting curve parameters of the equivalent circuit can be obtained by utilizing the fitting software, and the three parts of the impedance can then be obtained at the same time. The electrical resistance obtained under a high-frequency current is the sum of the solution impedance and membrane impedance R_m+s_. Similarly, to acquire the impedance of the membrane itself R_m_, it is necessary to conduct a blank experiment on the system without the membrane being tested and to thus obtain the solution impedance R_s_ [32].

## 3. Results and Discussion

Figure 2 illustrates the relationship between the membrane impedance and NaCl concentration, measured by chronopotentiometry. When the concentration of NaCl was greater than 0.2 M, the impedance did not change as the concentration changed. When the concentration was less than 0.2 M, the impedance values of both the cation- and the anion-exchange membranes, as well as that of the forward osmosis membrane, increased significantly as the concentration decreased. As a result, when the salt concentration was low, the change in membrane electrical resistance had a direct impact on the power density and power generation efficiency of the microbial fuel cell.

In order to further discuss the membrane electrical resistance at low concentrations, we studied the impedance of the membrane electrical resistance at different circulation velocities in 0.05 M (3 g/L) NaCl solution. Figure 3 shows the relationship between the membrane impedance obtained by DC chronopotentiometry and the solution flow rate at a low salt concentration. Under these circumstances, the measured impedance was much greater than that in the standard concentration (0.5 M) NaCl [33]. At low circulation velocities especially, when the circulation velocity decreased from 0.5 L/min to 0.3 L/min, the membrane electrical resistance increased rapidly. The cation-exchange membrane increased from 125 Ω cm^2^ to 461.3 Ω cm^2^ and the anion-exchange membrane increased from 160 Ω cm^2^ to 395 Ω cm^2^, while the positive osmosis membrane increased from 101 Ω cm^2^ to 270 Ω cm^2^, which, as revealed by Figure 3, led to a significant difference that could be seen in the three membrane electrical resistances, which may be due to the different ion mobilities in the different membranes [34].

Nevertheless, with the salt concentration mentioned above, the three membranes showed similar changing trends as the circulation velocity changed, and all decreased as the circulation velocity increased, while increasing the circulation velocity of solution obviously reduced the membrane electrical resistance. Therefore, in the case of a low salt concentration, the solution, rather than the membrane itself, played a decisive role in determining the membrane’s impedance. However, due to the limitations of DC chronopotentiometry, it can only be concluded that the circulation velocity can reduce the overall impedance, but we cannot distinguish which part of the impedance of the solution is influenced by the circulation velocity [35].

Figure 4 shows the impedance of the FO membrane and the cation- and anion-exchange membranes under high salt concentrations, utilizing the same approach. It can be seen from the figure that when the circulation velocity increased from 0.3 L/min to 0.5 L/min, the CEM membrane’s electrical resistance decreased from 22.3 Ω cm^2^ to 18.1 Ω cm^2^, the AEM membrane’s electrical resistance decreased from 13.5 Ω cm^2^ to 11.3 Ω cm^2^, and the FO membrane’s electrical resistance decreased from 10.2 Ω cm^2^ to 6.3 Ω cm^2^, which is much smaller than the change observed under a low salt concentration. As a result, the increase in the solution circulation flow rate cannot reliably reduce the membrane’s impedance, as it does under the condition of a low salt concentration [36]. Therefore, we can infer that, at a low concentration, the increase in the circulation velocity can reduce the overall impedance, which results from the reduction in the impedance of the boundary diffusion layer, while at high concentrations, the overall membrane impedance is affected by the impedance of the membrane itself, and different types of membrane impedance will result in obvious differences [37].

It can be seen from Table 1 that, when the concentration was 0.5 M and the circulation velocity was 0.4 L/min, the impedance of the FO membrane was 50% smaller than that of the CEM membrane, which may result from the fact that the fixed charge density on the surface of the FO membrane was less than that of the CEM membrane, leading to a reduction in the double layer impedance of the ions during membrane migration, and promoting the membrane delivery of the ions. Furthermore, another important factor contributing to the low impedance of the FO membrane is that the membrane was thin, at 52 μm. The CEM membrane usually requires higher mechanical strength; thus, it should be thicker, which also increases the membrane electrical resistance [38]. The impedance of the anion-exchange membrane (AEM) is higher than that of the FO, which may result from the different swelling ratios of the two membranes, under which circumstances different membrane characteristics are formed, such as membrane impedance, selective permeability, etc.

Figure 5 shows the changes in the membrane impedance of AEM, CEM, and FO as the circulation velocity changed at a solution concentration of 0.05 M. It is clear that the diffusion boundary layer impedance R_DBL_ contributed the most at each circulation velocity, regardless of the type of membrane and the circulation velocity solution, which is consistent with the results measured by the DC approach. When the circulation velocity of the solution was 0.1 L/min, the R_DBL_ values of the three membranes were 191 Ω cm^2^, 300 Ωcm^2^, and 130 Ω cm^2^, accounting for 76.5%, 81%, and 83% of the overall impedance, respectively. The true membrane impedance and double electric layer impedance of the three membranes contributed little to the overall impedance, accounting for a relatively minor part [39]. Furthermore, consistent with the DC approach, when the circulating flow rate was increased, the diffusion boundary layer impedance R_DBL_ decreased rapidly. When the circulation velocity was increased from 0.1 L/min to 0.8 L/min, the R_DBL_ decreased by 75%, 66.7%, and 80%, respectively. These results reveal that the membrane impedance can be reduced by increasing the solution’s circulation velocity at low salt concentrations, because the diffusion boundary layer’s impedance decreases noticeably as the solution circulation’s velocity increases. Figure 6, Figure 7 and Figure 8 shows the impedance of each part at the concentration of 0.05M NaCl, the experimental results indicate that the impedance R_M_ of the membrane itself and the electrical resistance R_DL_ of the double electric layer do not change significantly as the solution circulation velocity changes. Generally, the thickness of the double electric layer is on the nanoscale; therefore, the impedance values of the ions are not affected by the agitation of the solution when they pass through this layer [40]. By comparing the diffusion boundary layer impedance and the double electric layer impedance of the cation and anion membranes, it was found that the anion-exchange membrane is lower than the cation-exchange membrane, which may be related to the different ion mobility values of the chloride ion and sodium ion—u_Cl_/u_Na_ ≈ 1.5 [41].

Consequently, it can be seen from the abovementioned results that the impedance of the diffusion boundary layer is the main factor affecting the membrane at low salt concentrations, as shown in Figure 6, Figure 7 and Figure 8, and this part of the impedance can be reduced by increasing the solution’s circulation velocity, while the impedance values of the membrane itself and the double electric layer only contribute minimally [42].

Similarly, at a higher concentration, such as 0.5 M NaCl, the individual membrane impedance values of each of the three membranes were measured. Figure 9 reveals the relationship between the measured membrane impedance, the double electric layer impedance, the diffusion boundary layer impedance, and the circulation velocity. The results show that the membrane impedance was significantly higher than the diffusion boundary layer and double electric layer impedances, and the impedance fluctuated less as the circulation velocity changed. This differs from the results for the low concentration conditions, indicating that the membrane’s impedance is the main factor affecting the overall impedance [43]. Nevertheless, when the flow rate was 0.1 L/min, the CEM membrane’s impedance was 13 Ωcm^2^, accounting for 57.6% of the overall impedance. In this circumstance, the double electric layer impedance was only 2 Ωcm^2^, and the diffusion boundary layer impedance was 7.6 Ωcm^2^, accounting for 33.6% of the overall impedance. It can be seen that, at high salt concentrations, the diffusion boundary layer impedance still made a large contribution, but this part of the impedance decreased as the change in the circulation velocity increased. When the circulation velocity increased from 0.1 L/min to 0.8 L/min, the diffusion boundary layer impedance only accounted for 15.6% of the overall impedance. Furthermore, similar to the results derived at low salt concentrations, the impedance of the electric double layer did not decrease significantly as the change in the circulation velocity increased, as shown in Figure 10. One reason for this is that the thickness of the double electric layer is on the nanoscale, and the influence of the solution agitation is limited [44].

Therefore, the experimental results reveal that the diffusion boundary layer impedance is the main contributor to the membrane impedance at low salt concentrations, and its size can be reduced by increasing the circulation velocity of the solution. At high salt concentrations, the impedance of the membrane itself is regarded as the main contributor, but the boundary layer impedance should also be considered, as its contribution to the overall impedance is still significant [45].

The biggest difference between DC chronopotentiometry and AC impedance spectroscopy is that the latter can measure the overall impedance spectrum, quantify each sub-impedance by setting an equivalent circuit, and obtain the membrane impedance, solution impedance, double electric layer impedance, and diffusion boundary layer impedance. Consequently, the overall impedance values of the two test approaches should be equal [46], enabling a comparison of the overall impedance values obtained by the two test approaches, and it can be seen from the data that the overall internal impedance values obtained by the DC and AC impedance spectrum methods were equivalent. Figure 11 indicates that the reason for the higher mode impedance value given by the DC method at low concentrations is the rapid rise in the diffusion boundary layer impedance with the decrease in the concentration, instead of the increase in the membrane impedance. At low salt concentrations, the diffusion boundary layer impedance accounted for more than 76% of the total impedance, which explains why increasing the solution circulation velocity can reduce the impedance when applying the DC approach. In a high-salt concentration environment, the impedance of the membrane itself is the main contributor, and these results are consistent with the implications of the DC method [47]. This implies that the diffusion boundary layer impedance still makes a large contribution at high concentrations, as judged by the application of AC impedance spectroscopy.

Figure 12 demonstrates the relationship between the membrane impedance measured by the DC and AC approaches and the concentration of the external solution. It can be seen from the figure that there was a high correlation between the membrane impedance, as measured by DC or AC impedance spectroscopy, and the concentration of the external solution. When the concentration was lower than 0.5 M, the membrane impedance obtained by either DC or AC impedance spectroscopy decreased as the external solution’s concentration increased. When the C_low_ was above 0.5, the membrane impedance also decreased as the concentration increased, but this value generally tends to remain stable [48].

The free ion concentration (*C*_free_) in the membrane is closely related to the membrane’s impedance, which is defined as the difference between the equilibrium of the ion concentration and the fixed charge density of the membrane. Due to the electric neutrality of the membrane, the relationship between the fixed charge density *X* on the membrane (mol/m^3^ internal solution), the counter ions (with a fixed charge opposite to that of the membrane), and the co-ions is as follows:*X* + *C_co-ions_* = *C_counterions_*(2)

Therefore, this value depends on the concentration of co-ions, that is, *C*_*co-ions*_ = *C*_free_. According to existing research, based on the Donnan equilibrium, the relationship between the external salt solution concentration (*C_co-ions_*) and the co-ions concentration (*C_co_*_-*ions*_) is as follows [49]:(3)$$C_{co-ions}=\;\frac{1}{2}\left(\sqrt{X^{2}+{\left(2C_{\mathrm{ext}}\exp\left(-\mu^{*}\right)\right)}^{2}}-X\right)$$

It can be seen from the above formula that the external ion concentration C_ext_ is directly proportional to the concentration of co-ions *C_co-ions_*_._ As a result, the concentration of free ions in the membrane will change as the concentration of the external salt solution changes, and the size of the membrane’s impedance will also be affected.

Figure 13 demonstrates the relationship between the external salt solution conductivity, membrane conductivity, internal solution conductivity and external salt solution concentration (on the low concentration side). The results indicate that the external salt solution’s conductivity increased rapidly as the external salt solution’s concentration increased, and was always higher than the membrane conductivity [50]. The free ion conductivity in the membrane was equal to the membrane conductivity when the external solution concentration reached 0.73 M. Furthermore, when the external salt solution was less than 0.1 M, the membrane conductivity increased rapidly as the external salt solution concentration increased [51], and tended towards stability when it reached about 5 mS/cm. At this time, the external solution concentration continued to increase; that is, when it became higher than 0.3 M, although the internal solution conductivity C_free_ continued to rise, the membrane conductivity reached a threshold, and became stable in the region of 5.5 mS/cm.

## 4. Conclusions

In this paper, the Luggin capillary was utilized as part of DC chronopotentiometry (DC) and AC impedance spectroscopy (AC), and the membrane impedance was measured at low and high salt concentrations. As regards the DC approach, the decisive factor determining the membrane impedance under low salt concentration conditions was the solution, rather than the membrane itself. On the contrary, at high salt concentrations, the membrane impedance became the decisive factor, and different types resulted in different membrane impedances. Consistently with the DC approach, in the AC approach, under the same low salt concentration, the impedance values of the membrane itself and the double electric layer only contributed negligibly. The impedance of the diffusion boundary layer was the main contributor at low salt concentrations, and this part of the impedance could be reduced by increasing the flow velocity of the solution. At high salt concentrations, the membrane impedance was the main contributor, but the boundary layer impedance should also be considered, as its contribution to the overall impedance was still high. Through the correlation analysis, it was found that there was a high correlation between the membrane impedance values obtained by the DC and AC approaches and the external minimum concentration.

These results reveal that the membrane impedance obtained by both DC and AC approaches decreased as the lowest external solution concentration increased. When the low salt concentration was greater than 0.3 M, although the membrane impedance continued to decrease, the value tended gradually towards stability. Furthermore, the relationship between membrane conductivity and internal salt solution concentration was also investigated. When the external salt solution was less than 0.1 M, the membrane conductivity increased rapidly as the external salt solution concentration increased, and it tended towards stability at about 5 ms/cm. After this point, the membrane conductivity did not increase significantly and remained stable in the region of 5.55 mS/cm, which thus substantiates an approach to the further precise measurement and investigation of membrane impedance in an OsMFC system. It can be seen that the external ion concentration (C_ext_) was directly proportional to the free ion concentration (C_free_) in the membrane, and the free ion concentration C_free_ in the membrane was closely related to the membrane electrical resistance. As a result, a change in the concentration of the external salt solution will result in a change in the free ion concentration, and the membrane impedance will also be influenced.

## Figures and Tables

**Figure 1 membranes-12-00816-f001:**
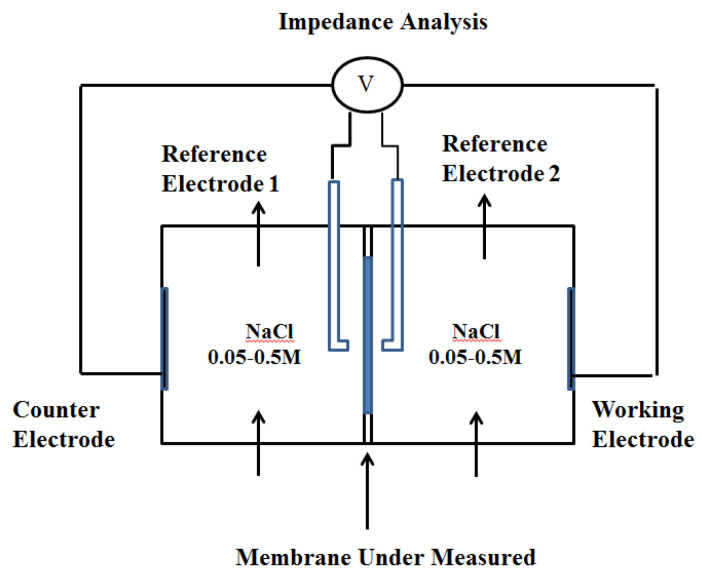
Schematic representation of the experimental setup.

**Figure 2 membranes-12-00816-f002:**
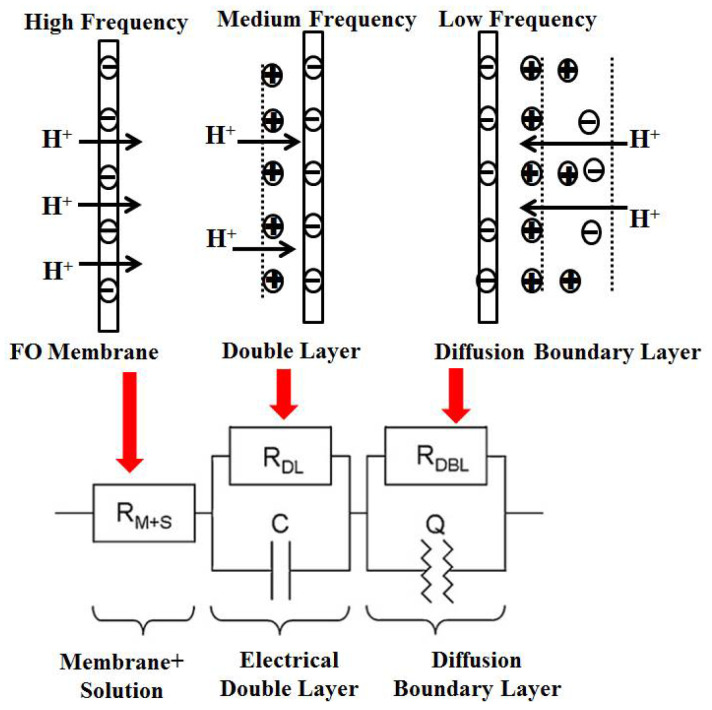
Partition of the membrane electrical resistance into equivalent electrical elements.

**Figure 3 membranes-12-00816-f003:**
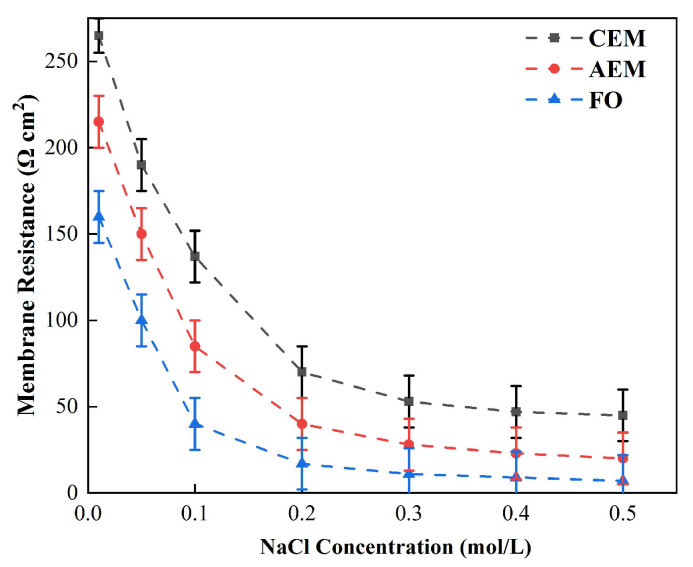
Relationship between the CEM, AEM, and FO membrane impedance and NaCl solution concentration. (The circulation flow rate of the solution was 0.5 L/min under the condition of DC.)

**Figure 4 membranes-12-00816-f004:**
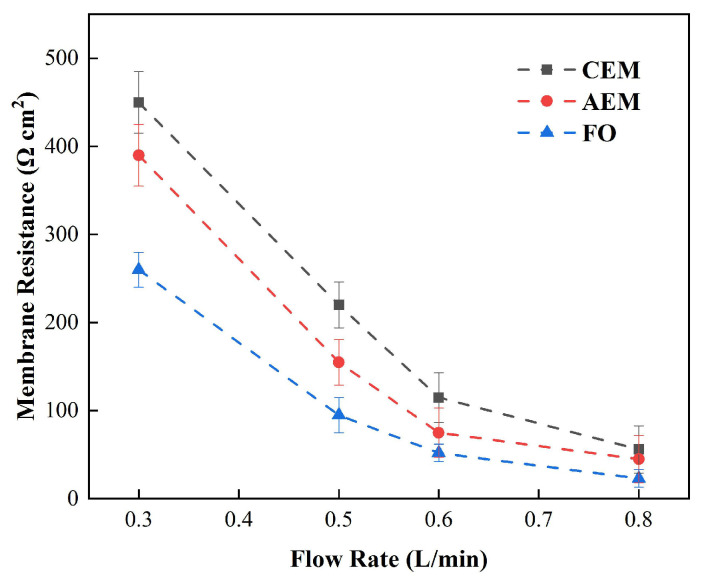
Relationship between the measured impedance and solution flow rate under the condition of DC (0.05 M NaCl).

**Figure 5 membranes-12-00816-f005:**
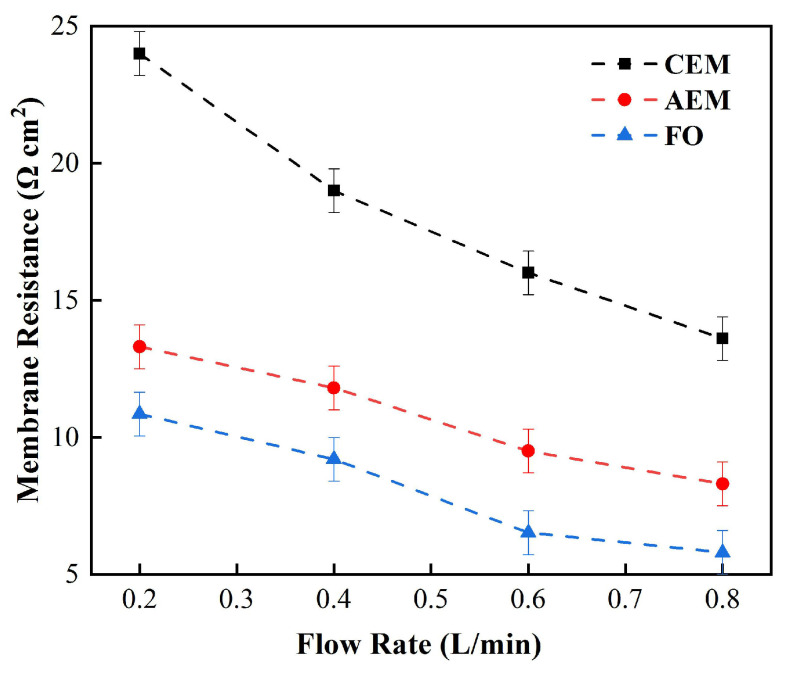
Relationship between the measured membrane electrical resistance and solution circulation velocity (0.5 M NaCl).

**Figure 6 membranes-12-00816-f006:**
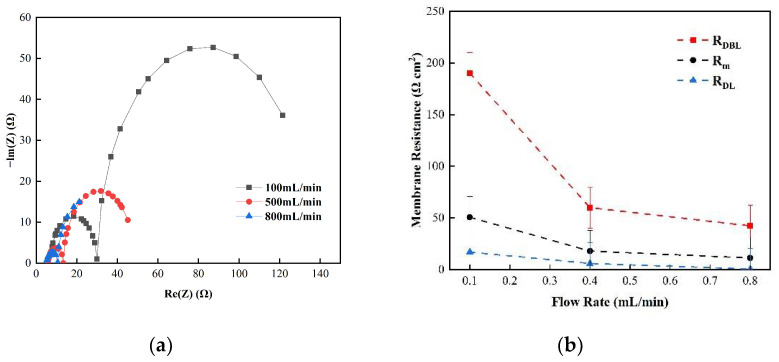
(**a**) The impedance of the Nyquist spectrum and (**b**) membrane impedance R_M_, double layer impedance R_DL_, and diffusion layer R_DBL_ of the AEM membrane in a 0.05 M NaCl solution.

**Figure 7 membranes-12-00816-f007:**
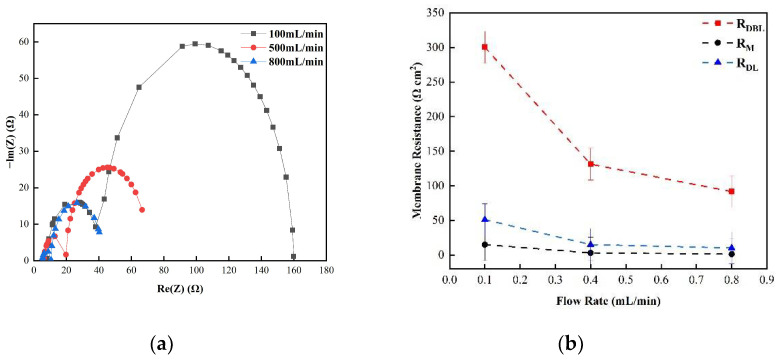
(**a**) The impedance of the Nyquist spectrum and (**b**) the membrane impedance (R_M_), double layer impedance (R_DL_), and diffusion layer impedance (R_DBL_) of the CEM membrane in 0.05 M NaCl solution.

**Figure 8 membranes-12-00816-f008:**
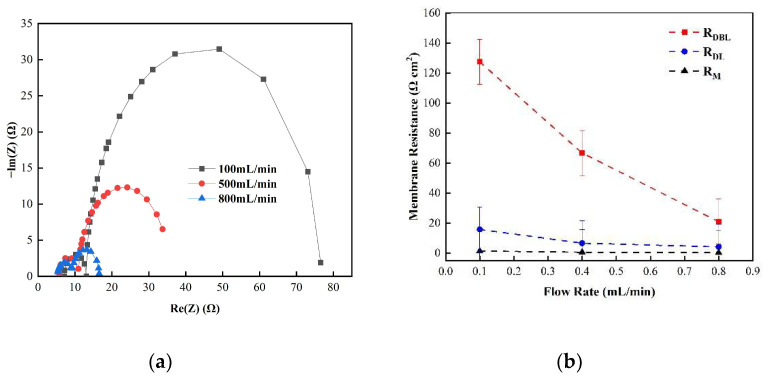
(**a**) The impedance of the Nyquist spectrum and (**b**) the membrane impedance R_M_, double layer impedance R_DL_, and diffusion layer impedance R_DBL_ of the FO membrane in 0.05 M NaCl solution.

**Figure 9 membranes-12-00816-f009:**
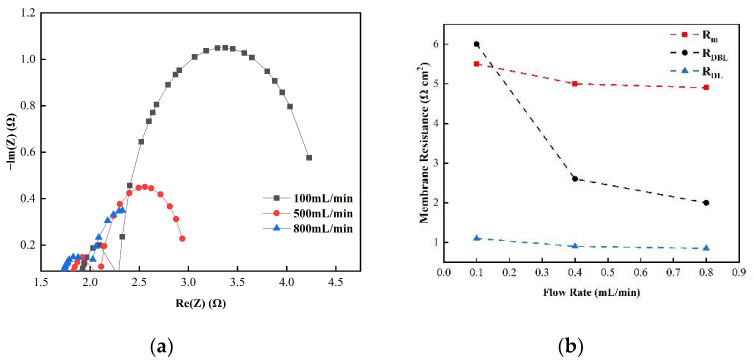
(**a**) The impedance of the Nyquist spectrum and (**b**) the membrane impedance R_M_, double layer impedance R_DL_, and diffusion layer impedance R_DBL_ of the AEM membrane in 0.5 M NaCl solution.

**Figure 10 membranes-12-00816-f010:**
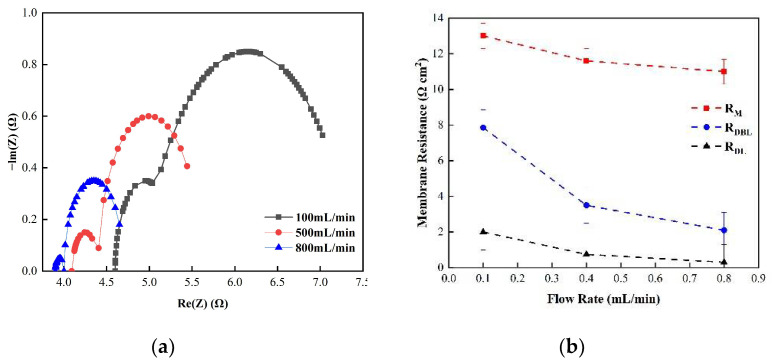
(**a**) The impedance of the Nyquist spectrum and (**b**) the membrane impedance R_M_, double layer impedance R_DL_, and diffusion layer impedance R_DBL_ of the CEM membrane in 0.5 M NaCl solution.

**Figure 11 membranes-12-00816-f011:**
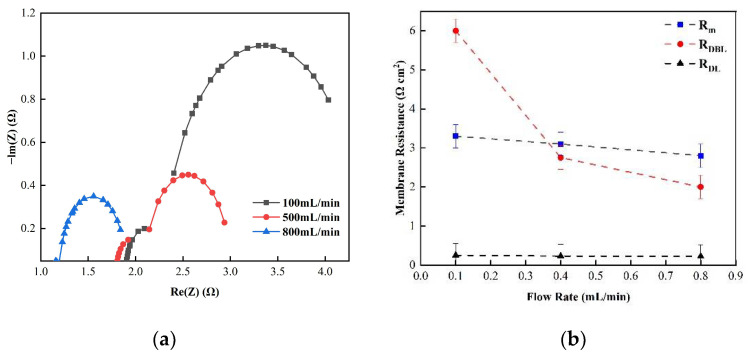
(**a**) The impedance of the Nyquist spectrum and (**b**) membrane the impedance(R_M_), double layer impedance (R_DL_), and diffusion layer impedance (R_DBL_) of the FO membrane in 0.5 M NaCl solution.

**Figure 12 membranes-12-00816-f012:**
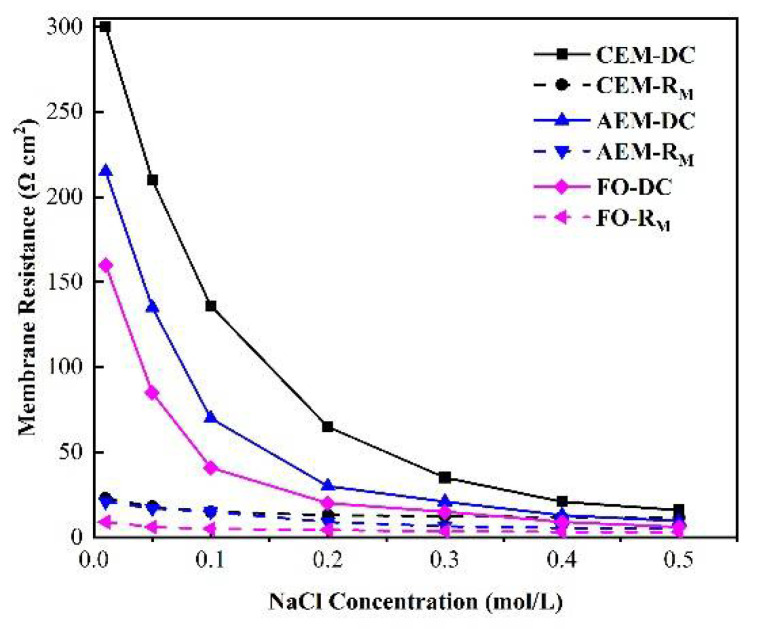
Relationship between the membrane impedance measured by the DC and AC approaches and the concentration of the external salt solution.

**Figure 13 membranes-12-00816-f013:**
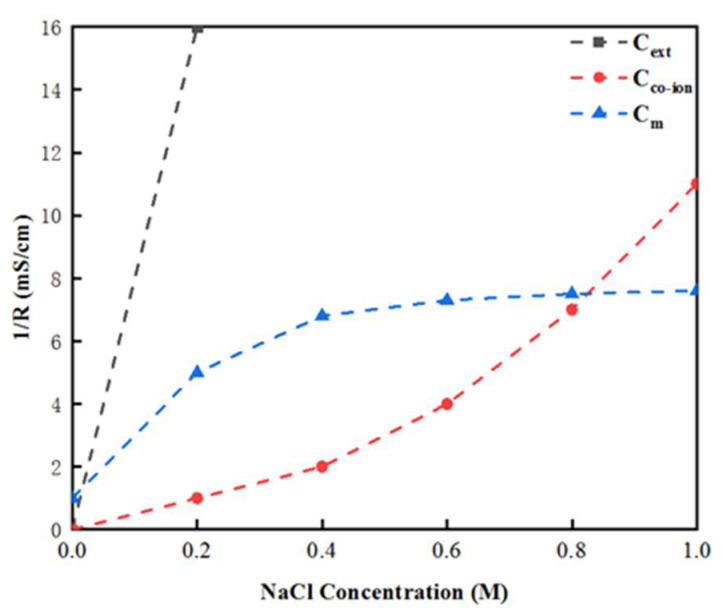
Relationship between the membrane conductivity measured by the AC approach, internal and external solution conductivity, and the concentration of the external salt solution.

**Table 1 membranes-12-00816-t001:** Test results of the membrane characteristics.

Type	Membrane Impedance ^a^ (Ω cm^2^)	Swelling Ratio ^b^ %	Thickness (μm)
AEM	12.2 ± 0.4	20.3 ± 0.6	82 ± 13
CEM	18.6 ± 1.6	21.5 ± 0.2	181 ± 2
FO	9.3 ± 0.4	38.1 ± 0.9	52 ± 18

^a^. The test conditions for a were 0.5 M NaCl solution at a temperature of 25 °C, and the circulation rate was 0.4 L/min. ^b^. The test conditions for b involved the measurement being performed after soaking in ultrapure water for 24 h.

## Data Availability

The data presented in this study are available on request from the corresponding author.

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
