# Peer review of "Research on Measuring Pure Membrane Electrical Resistance under the Effects of Salinity Gradients and Diffusion Boundary Layer and Double Layer Resistances"

_membranes, 2022, doi:10.3390/membranes12080816_

Round 1

Reviewer 1 Report

Hello,

Many thanks to the authors for considering the given comments. I believe the manuscript is now good for publication.

Regards

Author Response

Thank you very much for your support!

Reviewer 2 Report

1) The authors should supplement the introduction with a review of other studies on the impedance spectroscopy of ion exchange membranes.

2) The answer to the second comment should be reflected in the article

 3) Indicate in which works formula (1) is used

Reviewer 3 Report

The manuscript titled “Research on measuring the pure membrane resistance under the effect of salinity gradients, diffusion boundary layer and double layer resistances” and written by Yang Zhao and Duan Liang. Has still the same serious drawbacks. I recommend rejecting this study based on the following comments.

 1.       The purpose of this study is not well explained. I mean, what is the utility of obtaining the electrical resistance of membranes? Usually, the hydraulic resistance or permeability coefficients is determined through experimental runs to be used in the transport equations. This should be clarified by the authors in order to remark the aim of the study and what advantages has to have the electrical resistance parameter of the membranes. On this basis, the introduction should be extended.

22.       The reference section was not done appropriate. Check the format, references are not complete

Round 2

Reviewer 2 Report

Accept in present form

Author Response

Thank you very much for your support

Reviewer 3 Report

The manuscript titled “Research on measuring the pure membrane resistance under the effect of salinity gradients, diffusion boundary layer and double layer resistances” and written by Yang Zhao and Duan Liang could be interesting but, the authors have to improve the paper and clarify how useful is the determination of electrical resistance in membrane processes. I recommend a major revision based on the following comments.

1.       As it was already mentioned in the previous revision, usually, properties such us permeability coefficient or membrane resistance are measures in RO, FO, PRO, membranes and even the variation of this properties with the operating time and fouling. In this study, it was measured the electrical resistance, so this must be specified to avoid confusion. I mean, where “resistance” is written should be rewritten as “electrical resistance”

2.       In this line, the author should revise the determination of permeability coefficients and hydraulic resistance in membrane processes such as RO, FO and PRO by extending the introduction. And they should specify very clear the advantage of determining the electrical resistance over the permeability coefficient or hydraulic resistance and if the electrical resistance could be used for determining the fouling effect on these membrane processes. There are some studies that should be included related with permeability coefficient or hydraulic membrane resistance, I suggest the following:

a.       Water 11 (1), 152

b.       Journal of Membrane Science 444, pp. 523-538

c.       Environmental Science and Technology 46(8), pp. 4673-4681

d.       Desalination 533, 115768

e.       Journal of Membrane Science 476, pp. 410-420

f.        Computers & Chemical Engineering 153, 107441

g.       Journal of Membrane Science 554, pp.244-252

h.       Environmental Science and Technology 37(7), pp. 1432-1440

i.        Desalination 489,114526

j.        Desalination 136(1-3), pp. 281-286

k.       Processes 8 (6), 692

l.        Desalination 257(1-3), pp. 184-194

m.     Separation and Purification Technology 252,117455

n.       Desalination 397, 101-107

3.       Please, keep in mind that this sort of studies usually has between 35-60 reference due to the number of publications in last years.

4.       One more time, the reference section is not according with the style of the journal. I give one example of how a reference should be written:

a.       Qasim, M.; Badrelzaman, M.; Darwish, N.N.; Darwish, N.A.; Hilal, N. Reverse osmosis desalination: A state‐of‐the‐art review. Desalination 2019, 459, 59–104, doi:10.1016/J.DESAL.2019.02.008.

Round 3

Reviewer 3 Report

The authors have adressed all my comments

This manuscript is a resubmission of an earlier submission. The following is a list of the peer review reports and author responses from that submission.

Round 1

Reviewer 1 Report

The manuscript titled “Research on measuring the pure membrane resistance under the effect of salinity gradients, diffusion boundary layer and double layer resistances” and written by Yang Zhao and Duan Liang. has serious drawbacks. I recommend rejecting this study based on the following comments.

1.       The purpose of this study is not well explained. I mean, what is the utility of obtaining the electrical resistance of membranes? Usually, the hydraulic resistance or permeability coefficients is determined through experimental runs to be used in the transport equations. This should be clarified by the authors in order to remark the aim of the study and what advantages has to have the electrical resistance parameter of the membranes. On this basis, the introduction should be extended.

2.       The reference section was not done appropriate.

Reviewer 2 Report

1) The authors should add to the introduction an explicit description of the novelty of the work and provide a comparison with the approaches available in the literature for measuring resistances in membrane systems.

2) The sentence in lines 26-27 requires clarification and reference to the source.

3) The sentences in lines 31-32 and 43-46, 91-92 should also include references to sources.

4) In lines 97-99, it should be explained what type of dependence is obtained as a result of a chronopotentiometric study of the membrane and at what point in time the potential difference is fixed.

5) Check the designation "rdbl" in rows 118, 216.

6) The authors need to describe more clearly the algorithm for determining membrane impedance RM, double layer impedance RDL and diffusion layer RDBL.

Reviewer 3 Report

Hello,

In general, the article is tackling a very important and interesting topic, discussing the electrical resistance of the membrane as a critical component of many processes. However, some comments need to be considered to having the article more constructive and of high readership, which is summarized as follows:

1-         Careful proofreading is required (hopefully by an English linguist) to avoid many typos present and make the discussion more constructive.

2-         The introduction section should be improved with a more in-depth discussion on the importance of accurate measurement of membrane resistance in different processes. This has to be backed by relevant references and proper citations of relevant works.

3-         In line #77, the NaCl concentration is indicated to be 0.017-0.5 M, while it is indicated to be 0.05-0.5 M everywhere else (as in Fig. 1), please confirm on this.

4-         Some attention should be given to the units used and has to be revised. For example, concentration should be either in M or mol/l, as well as for circulation velocity of ml/min and L/min, so it should be unified!

5-         The complete experimental conditions should be listed in the caption of each figure, for example, the NaCl concentration for results in Fig. 4. Please indicate the differences between Fig. 4 and Fig. 5 in terms of conditions.

6-         Fix typo in line#229 for subscript and capital letters.

7-     In figures 6-11 indicate subfigures with (a), i.e. 6.a..etc. and identify each in the caption.

8-         Table 2 as referred to in line#288 is absent?!

9-         Indicate the title of “Conclusions”